# Social Aspects of Electric Vehicles Research—Trends and Relations to Sustainable Development Goals

**Vasja Omahne, Matjaz Knez and Matevz Obrecht ***

Faculty of Logistics, University of Maribor, Mariborska cesta 7, 3000 Celje, Slovenia;
vasja.omahne@student.um.si (V.O.); matjaz.knez@um.si (M.K.)
* Correspondence: matevz.obrecht@um.si

**Abstract:** E-mobility sustainability assessment is becoming more comprehensive with research integrating social aspects without focusing only on technical, economic, and/or environmental perspectives. The transportation sector is indeed one of the leading and most challenging greenhouse gas polluters, and e-mobility is seen as one of the potential solutions; however, a social perspective must be further investigated to improve the perception of and acceptance of electric vehicles. This could consequently lead to the European Green Deal's holy grail: faster decarbonization of the transportation sector. Another way to achieve it is by promoting more comprehensive sustainable development goals. Therefore, this paper combines a systematic review of recent research with research emphasis focused on social aspects of electric vehicles and their interconnection with specific UN Sustainable Development Goals. By knowing the current research focus mainly related with "perception" of electric vehicles and assessing their social "impact" as well as an emerging area of "user experience" and their relations with UN Sustainable Development Goals enables better insight on the current and future directions of electric vehicle social sustainability research. The current priority is identified as "climate actions". Increasingly important "sustainable cities and communities" shows potential for becoming one of the future research, policy, and community priorities.

**Keywords:** electric vehicles; social sustainability; social perspective; research trends; sustainable development goals (SDG)





## 1. Introduction

In 1987, the World Commission on Environment and Development presented the report "*Our Common Future*". They attempted to explain how important sustainability is to protect the environment for future generations and integrate social and economic progress. They also argued that governments should incorporate environmental considerations into decision-making [1]. The importance of sustainability was then expanded and was implemented in-laws, meaning countries started adopting sustainability-oriented laws [2], and at the same time, consumers became more and more aware of sustainability, mostly from the economic and environmental perspective [3]. This has also impacted the transportation sector, presenting considerable environmental, social, and economic challenges [4]. However, transport is vital from the economic perspective employing about 11 million people and generates almost 5% of the EU's GDP [5]. However, transportation accounted for about 24% of all greenhouse gas emissions in the EU. It consequently plays a significant role in air pollution resulting in climate change due to greenhouse gases (GHG) emissions [6].

Furthermore, road transport, in particular, accounted for about 18% of all EU emissions in 2012 [7]. A European Green Deal is one of the most impactful EU priorities that aims to transform the EU into modern, resource-efficient, climate-neutral, and sustainable. Decoupling growth from material use and emissions is especially challenging for sustainable mobility—a part of which is also e-mobility.

Replacing the internal combustion vehicles with new technologies such as electric vehicles (EV) could be a step towards more sustainable transport and reduced environmental impacts [6]. When just considering the use phase, reducing polluting emissions during driving of EVs is automatically achieved by all EVs [8] as they allow zero-emission driving. However, a negative impact is also perceived for using EVs since electricity for charging can be produced from environmentally disputable sources, e.g., fossil fuels [9]. Again, EVs impact the environment and human health in the production phase and end of life cycle, particularly particulate matter formation [10]. Furthermore, EVs' manufacturing presents a greater environmental burden with respect to gasoline cars, especially for the large use of metals, chemicals, and energy required by specific components of the electric powertrain such as the high-voltage battery [11]. Not only do EVs impact on the environment, but also on the social dimension. Onat et al. [12] argue that social impacts of alternative vehicle technologies should be further investigated to develop effective, sustainable mobility strategies. Furthermore, it is crucial to integrate the social perspective when studying electric vehicles' impacts, as the social perspective is interlinked with the environmental one [13]. Additionally, focusing only on the environmental perspective, substantial positive or negative social impacts regarding the electric vehicle impact can be overlooked; positive, for instance, presenting reduction of noise pollution, which can be positive or negative [14], while negative, for example, presenting the potential for exploitation of child labor [15]. Some of the impacts of EVs might even have social implications as byproducts (social impacts, social costs, etc.), which can influence EV acceptance and perception as well as social welfare. Some authors also expose user experience as one of the social factors as it is far from being related only to technical aspects. Moreover, 17 sustainable development goals (SDGs) defined by the United Nations (UN) to be achieved by 2030 represent a framework on which research and industry—also car manufacturers, as well as e-mobility in general—should focus in the future [16].

Inspired by the importance of social impacts of EVs, this paper presents a systematic overview of research papers related to EVs' social aspect in a five-year period of 2015–2019. With respect to reviewed papers, it was found out that linking EVs with their impact, user perception and acceptance was already researched. However, lacking examinations of their relation to UN SDGs was identified as well as dividing them into different categories.

This study is therefore focused on dividing papers among social factors of: (a) acceptance, (b) perception, (c) impact, (d) costs, (e) welfare, and (f) user experience, and it presents insight into research focus and trends on the expanding field of the social perspective of EVs from potential user perception, due to an identified lack of information on the correlation of all these factors with EV research papers. Moreover, reviewed papers are not specifying their relation to UN SDGs; therefore, this paper fulfills the gap identified in assessing relations of research papers with UN SDGs. Our main objectives are thus: (1) to define priority areas related to social sustainability from the user perspective; (2) to investigate research priorities related to the social perception of EVs in the field of (a) acceptance, (b) perception, (c) impact, (d) costs, (e) welfare and (f) user experience and; (3) to link reviewed papers with SDGs. Assessing SDGs priorities will enable interested researchers and organizations to see current and foresee future research and industry-related priorities of EVs related to specific SDGs.

## 2. Materials and Methods

The study is based on the systematic literature review approach proposed by the methodology developed by [17]. This paper provides a systematic and comprehensive review of available scientific papers to investigate the social perspective and social aspect of EVs. The papers and research for this study were based on Web of Science (WoS), and only review and scientific articles were considered, while book chapters and conference papers were excluded from this research since they are not strictly related to research papers, which focus on the SDG priorities are investigated in this paper. WoS databases are seen as one of the most often used and most distinguished scientific databases [18].

The time window considered was from 2015 to 2019 (5 years), as the paper selection procedure was conducted in November 2019 and to consider only papers containing results, which are "up-to-date." Papers were searched based on the different combinations of topics/keywords "electric vehicle" and "social impacts." The search comprehended 237 papers. When reviewing these papers as proposed in methodology [17], 65 papers were selected to be included in further analysis starting from reviewing titles and abstracts. After reviewing those papers carefully, 28 of them were picked, as those papers coincided with our scope, which was the evaluation of the impact of EV on society and human perception instead of focusing on environmental issues that can also be related to social impacts (e.g., noise pollution). They were further divided according to their evaluation of selected social factors as ranked in the framework developed in the study of preferences and needs of potential EV drivers on EV infrastructure [19]: (a) acceptance, (b) perception, (c) impact, (d) costs, (e) welfare and (f) user experience. Papers were sorted by the publication year, studied, and classified into the mentioned categories. The results were presented to provide a synthesis of findings with a special focus on research priorities among studied categories for separate studied years. Additionally, relation to 17 United Nations Sustainable Development Goals (SDGs) was examined for each paper [16], defining and identifying relations and prioritizing one or multiple SDGs. A tabelaric presentation of the results also gives a brief insight into the studied topics and the geographical focus and relation to the SDGs assigned.

## 3. Results

Priority Research Focus on EV Studies—Lessons Learned from a Systematic Literature Review

Papers published on the thematic of EVs' social impact have recently been frequently studied. The WoS research comprehended twenty-eight scientific papers published and related on topics "electric vehicle" and "social impacts" thematic and the graph that presents the publishing frequency per year can be seen in Figure 1. It can be seen that there has been an increase in the last two years of papers published regarding the social impact of EVs, meaning that this topic gains scientific as well as political importance. In 2018 and 2019, eight papers were published on the social impact of EVs thematic, respectively, while there were five papers published in 2017, four in 2016, and only three in 2015. This can also be supported by the fact that EVs are gaining more and more of the market share, especially in China [20], and this leads to EVs being studied more and more frequently. This is also due to methods, such as social life-cycle assessment (SLCA) gaining importance and being commonly used in the last three years [21] and due to social dimension and social sustainability being studied frequently regarding the supply chains [22].

As described in the methods section, we have grouped the papers into categories based on their contribution to the field of EVs' social impact. Figure 2 thus presents the grouping of papers, based on the categories, which are: "Acceptance", "Perception", "Welfare", "Social cost", "Impact" (using methods such as SLCA), "User experience" and "Readiness". The papers studied evaluated or studied mentioned categories regarding the EVs.

The authors studied the social aspect of "Acceptance", as of 28 papers, 4 coincided into the "Acceptance" category. Onat et al. [23] studied EVs' social acceptability, the environmental and economic impacts in the United States. Yousefi-Sahzabi et al. [24] also assessed the social acceptance of EVs in Turkey and concluded that the EVs are highly accepted and supported. Furthermore, the authors frequently focused on studying the "Perception" of EVs. For example, Brase [25] studied consumers' perception and decision-making about electric vehicles in the U.S., while Sovacool et al. [26] studied the perception that kids have of electric vehicles and sustainable transport in Denmark and the Netherlands. They [26] found out that children overwhelmingly seem to agree on the future direction of car-based transport, but cars must be safer, more energy-efficient, and alternatively fueled.

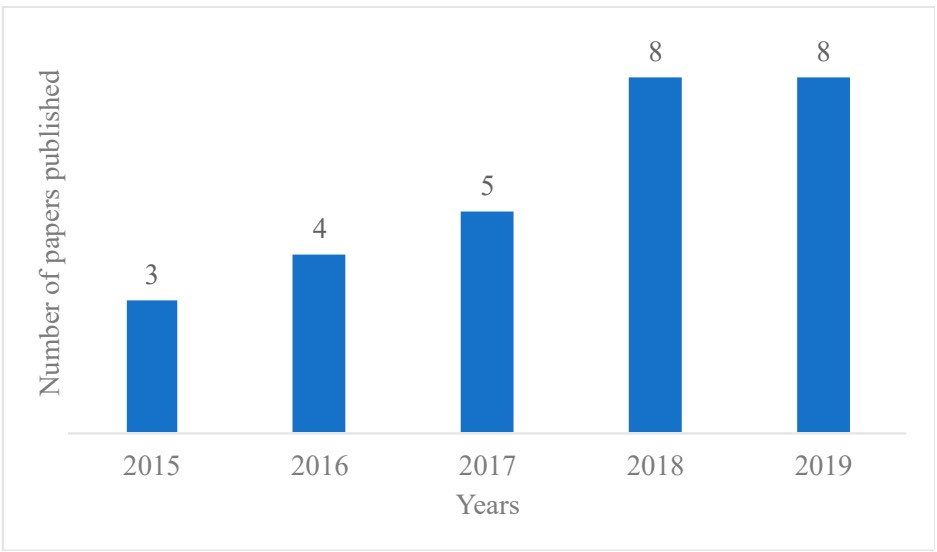

**Figure 1.** Published papers related to the social aspect of electric vehicles (EVs) (annual publications).

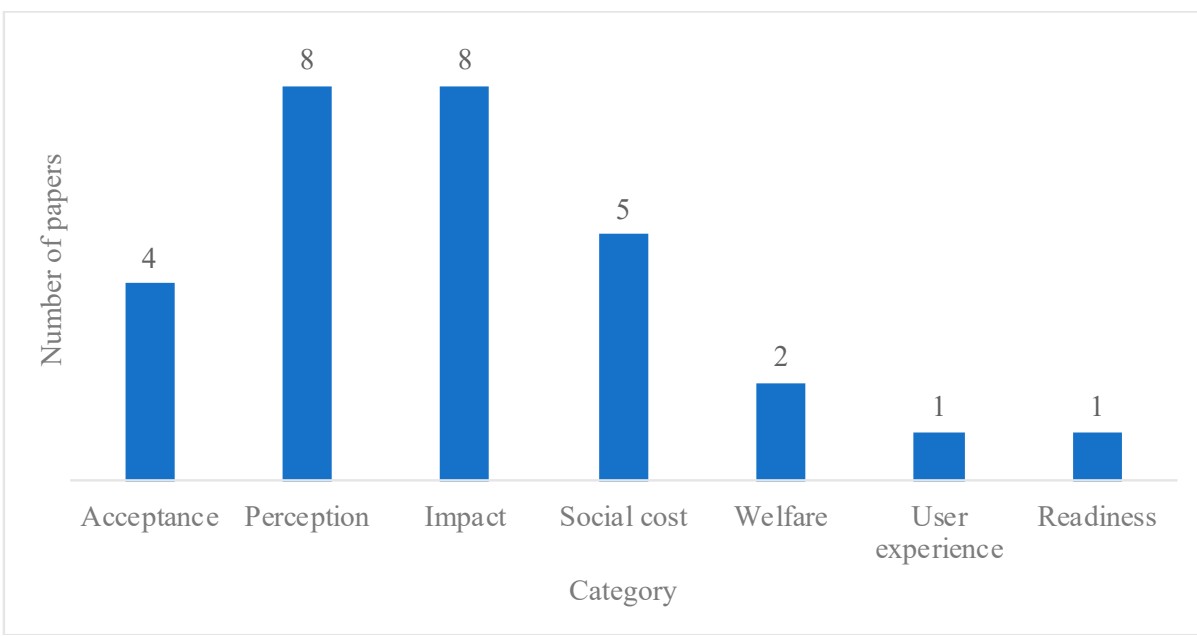

**Figure 2.** Papers divided in specific categories related to social aspects of EVs.

Investigating which topics are gaining or losing importance is highly important for defining future research priorities. Accordingly, the results in Figure 2 show the current research focus of studied papers divided into categories through the last five years per year, presenting which category has been given the central priority in a particular year.

Observing Figure 3, it can be seen that the year 2015 delivered papers that focused on perception, impact, and social cost of EVs. The number of impact papers has risen to two papers in 2016, while one paper related to social costs and one on the perception category was conducted in 2016. The year 2017 presented a reduction of impact studies, as none of the papers in the impact category have been published. Two papers related to category acceptance and one related to perception, welfare, and readiness respectively had been published in 2017. Furthermore, 2018 included three papers in the perception category, one in the acceptance category, two in the social cost category, and one in each impact and welfare category. The number of papers related to social impacts has once

again increased in 2019, as three papers on the social impact of EVs were published in 2019, and two perception related papers were also delivered. A user experience related paper was also published in 2019, showing that user as a research focus might be more critical as a research focus in the future.

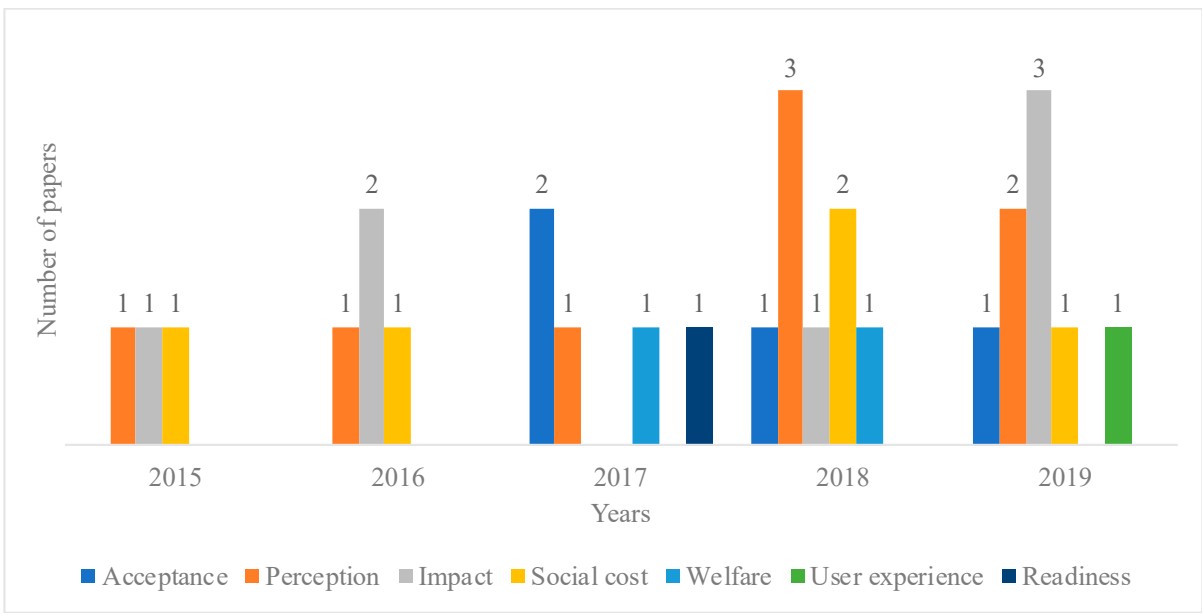

**Figure 3.** The research focus of papers related to social aspects of EVs included in Web of Science (WoS) in the last five years.

However, the authors allow the possibility that different social dimensions were studied and covered in published papers that were excluded from this study (e.g., papers published in journals not listed in WoS, conference papers, etc.

The authors most frequently focused on studying EVs' social impact with a method such as SLCA (or simplified SLCA). Social impact assessment of EV was identified in 8 out of 28 papers. For instance, Onat et al. [27] used seven sustainability indicators to indicate the social, environmental, and economic impact of different vehicles, including EV and perceived EV, to be the best alternative in the long-term for reducing human health impacts and air pollution from transportation. Onat et al. [28] also evaluated the social impact of EVs, although employing a life-cycle sustainability assessment, which includes the SLCA. They also presented a framework for assessing the sustainability of EV. They perceived that the optimal vehicle distribution in the U.S., considering the socio-economic indicators, would comprise internal combustion vehicles in the majority. Albergaria de Mello Bandeira et al. [29] also assessed EVs' sustainability in the last mile delivery and perceived several positive social impacts of EVs.

EVs social costs were investigated in 5 out of 28 papers. Newbery and Strbac [30]) evaluated the EVs regarding what is needed for battery electric vehicles to become socially cost-competitive, while Luo et al. [31] analyzed charging stations for EVs to minimize social costs, which are in both cases bigger than conventional vehicles.

Only two papers covered category welfare (one user experience and one readiness), meaning these categories were less addressed, but user experience might become more critical in the future.

The distribution of the published papers among journals has shown that most of the papers related to social aspects of EVs were published in the journal Transportation Research Part D (Transport and Environment), followed by Applied Energy and Transportation Research Part A (Policy and Practice) (Table 1).

**Table 1.** List of journals covering the social aspect of electric vehicles (EVs).

| | |
|---|---|
| Transportation Research Part D (Transport and Environment), | 6 |
| Applied energy | 3 |
| Transportation Research Part A (Policy and Practice) | 3 |
| Technological Forecasting & Social Change | 2 |
| Energy | 2 |
| Journal of Cleaner Production | 2 |
| Renewable and Sustainable Energy Reviews | 2 |
| Other journals | 8 |

Papers included in the detailed analysis are presented in Table 2. Additionally, the geographical area of the conducted study for each paper is also shown. Moreover, the studied paper was related to 17 United Nations Sustainable Development Goals (SDGs) [16].

**Table 2.** Research focus, category, geographical area, and relation to the sustainable development goals (SDGs) (alphabetically).

| Paper | Paper Focus | Social Aspect Category | Geographical Area | Relation to SDGs |
|---|---|---|---|---|
| Albergaria de Mello Bandeira et al. (2019) [29] | Proposing a method to assess alternative strategies for the last-mile of parcel deliveries in terms of social, environmental, and economic impacts | Impact | Brazil (Rio de Janeiro) | Climate action, Sustainable cities and communities, Decent work and economic growth, Good health and well-being |
| Andwari et al. (2017) [32] | Evaluating the technological readiness of the different elements of BEV technology | Readiness | - | Climate action, Sustainable cities and communities, Decent work, and economic growth |
| Brase (2018) [25] | Psychology of consumer perceptions and decisions about EVs | Perception | U.S.A | Climate action, Sustainable cities and communities, Decent work, and economic growth |
| Cherchi (2017) [33] | Measure the effect of both informational and normative conformity in the preference for EV | Perception | - | Sustainable cities and communities |
| Daramy-Williams et al. (2019) [34] | Reviewing the user experience, driving EVs | User experience | UK | It cannot be defined |
| Fang et al. (2018) [35] | Estimating marginal emission rates of electricity and the marginal price of electricity provided for charging EVs at different times of the day | Social cost | U.S.A. | Climate action, Sustainable cities, and communities, Decent work and economic growth |
| Giordano et al. (2018) [36] | Comparing diesel and battery electric delivery vans from an environmental and economic perspective | Impact | EU, U.S.A. | Climate action, Sustainable cities, and communities, Decent work and economic growth, Good health and well-being |
| Günther et al. (2015) [37] | The study analyzes where jobs could be created or cut down and the other two dimensions of sustainability | Impact | Germany, China, EU | Climate action, Sustainable cities, and communities, Decent work and economic growth |
| Hardman et al. (2016) [38] | The distinction between high-end adopters and low-end adopters | Perception | - | Climate action, Sustainable cities, and communities, Decent work and economic growth |
| Helveston et al. (2015) [39] | Consumer preferences for conventional, hybrid electric, plug-in hybrid electric (PHEV), and battery electric (BEV) vehicle technologies | Perception | China, U.S.A. | Climate action, Sustainable cities and communities, Decent work and economic growth |
| Herrenkind et al. (2019) [40] | Conducting qualitative research to identify relevantly factors influencing individual acceptance of autonomously driven electric buses | Acceptance | Germany | Sustainable cities and communities, Industry, innovation, and infrastructure |

**Table 2.** *Cont.*

| Paper | Paper Focus | Social Aspect Category | Geographical Area | Relation to SDGs |
|---|---|---|---|---|
| Kershaw et al. (2018) [41] | Assessing the contemporary 'consumption' of the motor-car in the context of increased uptake of EVs as part of a transition to a low carbon automobility | Perception | UK | Climate action, Sustainable cities and communities, Decent work and economic growth |
| King et al. (2019) [42] | Investigating the effects of stereotype threat on EV drivers | Perception | UK | Sustainable cities and communities |
| Kontou et al. (2015) [43] | Optimal electric driving range of (PHEVs) that minimizes the daily cost borne by the society when using this technology | Social cost | U.S.A. | Climate action, Sustainable cities, and communities, Decent work and economic growth, Industry, innovation and infrastructure |
| Luo et al. (2018) [31] | Proposing an optimization model for minimizing the annualized social cost of the whole EV charging system | Social cost | China | Sustainable cities and communities, Decent work and economic growth, Industry, innovation, and infrastructure |
| Luo et al. (2019) [44] | Proposing a comprehensive optimization model concerning the joint planning of distributed generators and EVs charging stations | Social cost | China | Climate action, Sustainable cities and communities, Decent work and economic growth, Industry, innovation and infrastructure |
| Newbery & Strbac (2016) [30] | What would make EVs to become socially cost competitive | Social cost | UK | Sustainable cities and communities, Decent work and economic growth |
| Onat et al. (2016a) [27] | Uncertainty-embedded dynamic life cycle sustainability assessment framework | Impact | U.S.A. | Climate action, Sustainable cities and communities, Decent work and economic growth, Good health and well-being |
| Onat et al. (2016b) [28] | To advance the existing sustainability assessment framework for alternative passenger cars | Impact | U.S.A. | Climate action, Sustainable cities and communities, Decent work and economic growth, Good health and well-being |
| Onat et al. (2017) [23] | Suitability of battery electric vehicles in the United States and the social acceptability of the technology | Acceptance | U.S.A. | Climate action, Sustainable cities and communities, Decent work and economic growth, Affordable and clean energy |
| Onat et al. (2019) [12] | Presenting a novel comprehensive life cycle sustainability assessment for four different support utility EV technologies | Impact | Qatar | Climate action, Sustainable cities and communities, Decent work and economic growth, Good health and well-being, Clean water and sanitation |
| Pautasso et al. (2019) [45] | Proposing a model for evaluating environmental, social, and economic impacts exerted by the diffusion of EVs | Impact | Italy | Climate action, Sustainable cities, and communities, Decent work and economic growth, Good health and well-being |
| Shao et al. (2017) [46] | Social welfare of monopoly and duopoly of EVs and gasoline cars | Welfare | - | Climate action, Sustainable cities and communities |
| Sovacool et al. (2018) [47] | Assessing of the demographics of electric mobility and stated preferences for EV | Perception | Nordic region | Climate action, Sustainable cities and communities, Decent work and economic growth |
| Sovacool et al. (2019) [26] | Assessing how schoolchildren between 9 and 13 years of age think about electric mobility | Perception | Denmark, Netherlands | Sustainable cities and communities |
| Wang et al. (2019) [48] | Explore the potential factors that affect consumers' acceptance of EVs in Shanghai | Acceptance | Shanghai | Climate action, Sustainable cities, and communities |

**Table 2.** *Cont.*

| Paper | Paper Focus | Social Aspect Category | Geographical Area | Relation to SDGs |
|---|---|---|---|---|
| Yousefi-Sahzabi et al. (2017) [24] | Social acceptance of low-carbon energy technologies in Turkey and the current status of the energy sector from a social perspective related to EVs | Acceptance | Turkey | Climate action, Sustainable cities, and communities, Decent work and economic growth, Affordable and clean energy |
| Zheng et al. (2018) [49] | Investigating the impact of EV manufacturing- and society-related factors on balance among manufacturer profits, environmental impact, and social welfare. | Welfare | China | Climate action, Sustainable cities, and communities, Decent work and economic growth |

Table 2 presents that most of the studied papers are national studies related to one country or even region only (20 papers). Only four included more than one country and could be identified as international studies. The remaining four papers did not specify their geographical orientation.

Table 2 and Figure 4 display that most papers contribute or are related to one or more UN SDGs. Most identified SDG was the relation with "sustainable cities and communities" (96.43% of papers are related to this goal), "climate action" (75% of papers are related to this goal), and "decent work and economic growth" (75% of papers are related to this goal). It is seen in Figure 4 that "sustainable cities and communities" was one of the priorities through all studied years but rocketed in 2019 to be much more represented than other SDGs.

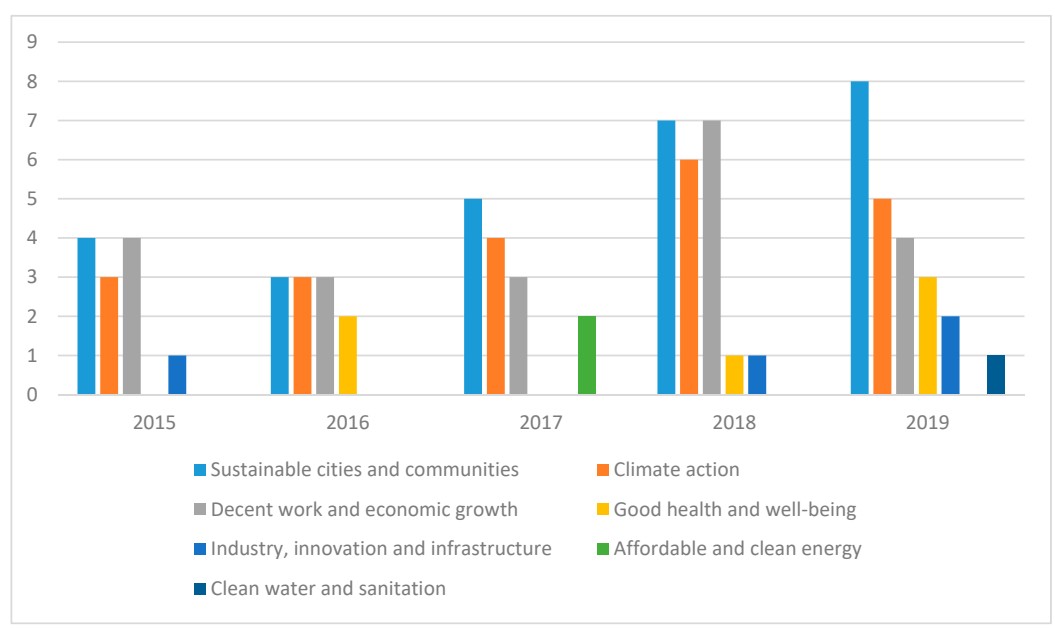

**Figure 4.** Relation and focus of studied papers towards UN sustainable development goals.

Papers are less related to SDGs such as "affordable and clean energy" (only 3.57% of papers related), "good health and well-being" (17.86% of papers related), "clean water and sanitation" (only 3.57% of papers related). "industry, innovation and infrastructure" (14.29% of papers related) seem to have moderate importance, and research focus in some cases can be associated with this goal as well, mainly due to investigating charging infrastructure associated with EVs.

Almost all papers are related to more than one SDG. The study of Albergaria de Mello Bandeira et al. [29] was related to goals "climate action", "sustainable cities and communities", "decent work and economic growth," and "good health and well-being". As the authors focused on proposing a method to assess alternative strategies for the

last-mile of parcel deliveries, they also focused on the impact on worker's health, and the paper is therefore related to "good health and well-being". The goal is to ensure healthy lives, and this can be done by first evaluating and then reducing the impact of different factors on worker's health. Relation with "climate action" can also be identified since it assesses the environmental impacts of EV use and it takes urgent action to combat climate change and its impacts. Albergaria de Mello Bandeira et al. [29] study further relates to "decent work and economic growth", which aims to promote sustained, inclusive, and sustainable economic growth. As they assess different delivery strategies from an economic perspective, they support sustainable economic development. The paper also relates to "sustainable cities and communities", the goal of which is to make cities and human settlements inclusive, safe, resilient, and sustainable. They assess postal deliveries in Rio de Janeiro from all three sustainability perspectives to make cities safe and sustainable.

## 4. Discussion

As it was comprehended from the analysis of published literature, it can be argued that there is a lack of papers that evaluate the impact of EVs regarding social welfare and user experience, and it presents a significant challenge to be addressed. Lack of papers was also noted in the categories of welfare and user experience. However, acceptance and perception is frequently studied, although there is a lack of incorporation of studying EVs image, regarding the perception of them as a status symbol. Furthermore, those papers rarely addressed the EV's impact on helping individuals to gain new friendships. Additionally, papers evaluating the social impact frequently studied economic and environmental aspects besides social aspect, which presents excellent progress regarding the sustainability evaluation and a trend, which is the evaluation of all three dimensions of sustainability.

For example, papers should combine the mentioned user experience and social welfare evaluation and evaluate one perspective, which is user experience, and evaluate social welfare and make the study more comprehensive. Furthermore, no papers have combined all the categories when evaluating EVs' social aspects. This can present a challenge for future studies and can be perceived as a gap needed to be addressed in the future. For future studies, it is also important to focus more on society's readiness level, as this is also a less addressed category and should be integrated into evaluating the perception or acceptance to boost EV commercialization. Society's readiness level is linked with approval and also with their perception of EVs. However, it is interesting that when authors tried to evaluate EVs' social aspect, they frequently also assessed an environmental and economic aspect of sustainability. It was found out that 87.5% of the papers assess the social impacts of EVs also integrated economic and environmental aspects. This is because all three aspects of sustainability are interlinked [13] and because studying sustainability comprehensively presents the studies' additional added value. This shows a trend and also potential future focus of studies regarding the evaluation of the social aspect of EVs, as more and more companies (e.g., car manufacturers, distributors, wholesalers, battery producers, software developers for self-driving vehicles) want to integrate all three dimensions of sustainability into their processes [50].

Considering the years of published papers, EVs' perception is gaining importance, as the number of papers, focusing on EV perception, has been increasing, especially in 2018 and 2019. Although the number of perception related papers on EVs could potentially decline, as the actual functionality of the technology improves, but also as consumers are exposed through the social, industry, and policy-related channels to more information about the technology and how it works [51]. Additionally, the number of EV papers on social impact has fallen from 2016 to 2017 but has been rising since 2018. This could be because the number of EVs has been snowballing since 2017 with annual increases above 50% [52,53]. As a vast number of consumers are gradually adopting the technology, companies try to optimize their processes while evaluating sustainability, which leads to substantial financial and environmental benefits [54,55]. Countries are also adopting sustainability-oriented laws [2], which impact mentioned companies to be more sustainable

and evaluate the sustainability [55] as well as linking EV sustainability with energy for charging them [56]. Greater focus is also dragged by novelties in batteries and charging systems, emphasizing the life cycle assessment of batteries [57]. Even though significant advance is noted in technology development, we believe that the social aspect is equally essential for greater EV commercialization and should be carefully examined for different cultures and countries. Most of the published papers were geographically focused on one country only. To get a better insight into the potential differences among cultures and countries studies including multiple countries with an international scope, as the study of Knez et al. [53] on evaluating differences among the different EU Member States with similar GDP or population density would be beneficial for improved understanding of EV related social factors as well.

Therefore, countries and organizations related to EV development, production, distribution, sales, use and end-of-life are trying to achieve sustainability-related goals that can be easily identified in well-categorized UN SDGs. Currently, papers were most frequently related to SDGs "sustainable cities and communities" followed by "climate action" and "decent work and economic growth". This could be because EVs present an option to facilitate climate action by providing emission-free transport options [8] and can present positive effects on cities, which can become more sustainable. Papers are rarely related to "good health and well-being", "industry, innovation and infrastructure," and "affordable and clean energy". However, it is interesting that the papers were frequently related with "climate action" and rarely with "affordable and clean energy," which can be highly interlinked, especially in the case of EV charging that can be sustainable and climate-friendly if electricity is produced from carbon-neutral electricity mix. More papers were related to "climate action" and "decent work and economic growth," including evaluating the environmental and economic impacts of EVs besides social implications. Future research on assessing the EV's social impact should also focus on developing numerical indicators such as the lowest-paid worker, the minimum wage, a number of holidays effectively used by employees etc. [15]. Additionally, research should use more indicators considering the social impact. Papers that studied social impact, in general, did not use a large number of indicators. However, the authors allow the possibility that different social dimensions were studied and covered in published papers that were excluded from this study (e.g., papers published in journals not listed in WoS, conference papers).

To conclude, this research provides new insight and reveals EVs social sustainability research focus defined and structured by studying papers included in WoS published in the last five years. The research comprehended a lack of papers related to user experience, social readiness, and welfare. Further, it also suggested that evaluating the social impact of EV, environmental, and economic aspects is also frequently addressed and integrated into the evaluation, making papers more comprehensive and presenting a smart solution. Future research should develop numerical indicators when evaluating EVs' social aspect and use a variety of different indicators. It might be considered in comprehensive measuring of research and business activities related to strategic and operational goals that can be directly or indirectly linked with achieving the UN SDGs.

**Author Contributions:** Conceptualization, M.O. and M.K.; methodology, validation and writing—original draft preparation, visualization, V.O. and M.O.; formal analysis and investigation V.O.; writing—review and editing; supervision, M.O. and M.K.; project administration, M.O. All authors have read and agreed to the published version of the manuscript.

**Funding:** This research received no external funding.

**Conflicts of Interest:** The authors declare no conflict of interest.

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
