# Peer review of "Social Aspects of Electric Vehicles Research—Trends and Relations to Sustainable Development Goals"

_wevj, doi:10.3390/wevj12010015_

Round 1
Reviewer 1 Report
In Introduction, authors should add a sentence to connect World Commission Report and situation in EU. Now there is no link with the World Commission requirements and EU trends in transportation sector.
There is grammatical mistake in row 46 and 47.
In Chapter 2., maybe the list of journals that publish 28 papers related to the paper. Why the rest of 237 papers are not examined and what is their scope in EV research. It is not obligatory, just an opinion.
Figure 2. studies or papers?
Sentence from row 126 to 130 is too long and can confuse the reader.
Row 147, sentence can't start with number and needs to cleared for the whole paper. Sometimes authors used word studies and sometimes papers. I know this are the studies that are converted to papers, but this needs to be written in cleared way. Something like...authors make study on...or...in paper by authors... research on social impact is conducted...
Author Response
The authors would first like to thank the reviewer for the thorough review, comments, and improvement proposals.
Long sentences were shortened to be more concise. Spelling mistakes were corrected (e.g. line 46-47) and the paper was additionally proofread by a native speaker.
Thank you for proposing to link the WCED report to EU priorities – it is now included in the introduction section. Current EU priority – European green deal is also presented and further coupled with sustainable mobility, a part of which is also e-mobility.
The goal of the paper is better explained and clarified additionally.
In Figure 3 “studies” was replaced by “papers” as proposed.
Studies and papers were replaced by “papers” since our research focus were published papers and as you mentioned – not actual studies, that may be published as more related or not related scientific or professional papers or maybe even not get published at all.
The process of paper selection within the review study is also clarified and explained with an explanation of selected categories of focus.
A list of journals was added as well, and some comments regarding the geographical orientation of papers with a focus mainly on national studies, which should be upgraded to more international studies comparing differences among different countries, cultures,…
Reviewer 2 Report
The paper investigates the progress of literature with respect to social aspects of EVs. While the topic is interesting and could definitely be of value to the academic community, I have some major concerns (listed below) regarding the current status of the paper.
My main issue is that the paper is not clear about its goals, the way it defines concepts and selects the examined data set. Furthermore, the claims in the conclusions are not refined enough to reflect the analysis conducted in the paper. Instead, they might be generic or misleading. I offer detailed suggestions and comments below.
Major comments
- What do you mean with the first sentence in the abstract? Do existing papers focus on the social aspect of EVs but no on technical or socio-economical factors?
- On page 2, line 68-69 the authors refer to the “social perspective”. The term is very broad and could include a lot of aspects, therefore, the authors should define what they exactly mean by social perspective. Some of the impact of EVs might have social implications as biproducts. Is such an impact part of the focus of this work?
- The paper states: “The search comprehended 237 papers, from which 28 were selected, as those papers coincided into our scope, which was the evaluation of social impact of EVs.” Based on which criteria only 28 papers were kept? What made the rest 200+ papers not suitable for this commentary? It is important to clarify this point because the remaining 28 papers do not make a sufficiently large sample to make strong claims. Also, it is very surprising that only 28 papers in the past 5 years dealt with the social aspects of EVs, considering the popularity of the topic. Thus, first the term “social perspective” needs clear definition and second, the exact criteria for paper selection need to be articulated.
- Somewhat related to my previous comment, the paper states “It can be seen that there has been a significant increase in the last two years”. However, an increase of 5 to 8 papers in a sample of 28 is approx. a 10% increase. Can such an increase be claimed as significant in such small numbers?
- Why did you select further categories as “Acceptance”, “Perception”, “Welfare”, “Social cost”, “Impact”. What is the rationale behind this grouping?
- Also, adding to the previous comment, looking into the EV literature, without conditioning your search on the “Social perspective” you would have found many more articles dealing with “impact”, “social welfare”, etc. So, Figure 3, which indeed can be informative, would have been more populated, and allowing for making stronger claims.
- The paper states that “Observing Figure 3, it can be argued that year 2015 comprehended only studies that focused on perception, impact and social cost of EVs.”. I am not convinced given that this claim is only based on 3 papers.
- As a result of the limited data set of the article the first claim of the discussion is flawed “it can be argued that there is a lack of studies that evaluate the impact of EVs regarding social welfare and user experience and it presents a great challenge to be addressed” There are plenty of articles dealing with social welfare and EVs. Perhaps fewer articles are dealing with user experience, however, the way the data set is conditioned leads to wrong conclusions. The authors should clarify/adjust their conclusions or expand their data set.
Minor Comments
- The paper requires a very good English proofreading. Currently, it is hard to understand. There are articles missing and many other linguistic mistakes, i.e., it is “The transportation sector..”, “the social perspective...”, “WoS databases are seen..."
- Also there are a lot of logical jumps without the respective explanations.
In summary, the paper can have a lot of potential, however, the sloppy writing, together with the missing elements with regards to definitions, sample selection and general analysis, make it hard for the reader to appreciate such potential.
Author Response
The authors would first like to thank the reviewer for the thorough review, comments, and improvement proposals.
The paper was additionally proofread by a native speaker and spelling mistakes were corrected (e.g. line 46-47). Long sentences were shortened to be more concise and easy to read.
Current EU priority – European green deal is also presented and further linked with sustainable mobility, a part of which is also e-mobility.
The goal of the paper is better explained and clarified additionally. The process of paper selection within the review study is also further described and explained in Methods to explain selected categories of our paper’s focus.
Words “studies” and “papers” were replaced by “papers” only since our research focus were published papers and as you mentioned – not actual studies, that may be published as more related or not related scientific or professional papers or maybe even not get published at all.
A list of journals was added as well, and some comments regarding the geographical orientation of papers with a focus mainly on national studies, which should be upgraded to more international studies comparing differences among different countries, cultures,…
The first sentence of the abstract is there to expose that studies were recently more focused on technical (e.g. range, power, charging), economic (economics, costs, etc.) and environmental perspective (e.g. carbon footprint, environmental assessment based on LCA approach etc.) but studies that focus specifically on social aspect are not that common. No matter that, the first sentence and the whole abstract were modified to be more precise and more concise.
The social perspective / aspect and focus of the paper is additionally explained. “This study is focused on social: a) acceptance, b) perception, c) impact, d) costs, e) welfare and f) recently also user experience” as those papers coincided with our scope, which was the evaluation of the impact of EV on society and human perception instead of focusing on environmental issues that can also be related to social impacts (e.g., noise pollution). Papers were further divided according to their evaluation of selected social factors as ranked in the framework developed in the study of preferences and needs of potential EV drivers on EV infrastructure [19]: a) acceptance, b) perception, c) impact, d) costs, e) welfare and f) user experience.
It was included in the review that the impacts of EVs might even have social implications as by products (social impacts, social costs etc.). This is added in the introduction with the explanation that social implications as by-products can be influential when examining EV acceptance and perception as well as social welfare. Some authors also expose user experience is one of the social factors since it is far from being related only to technical aspects.
Regarding the statement “It can be seen that there has been a significant increase in the last two years” for an increase from 5 to 8 papers, we agree entirely that “significant” is an inappropriate word choice and was therefore deleted.
Regarding your comment on: “Observing Figure 3, it can be argued that the year 2015 comprehended only studies that focused on perception, impact and social cost of EVs.” It was changed to: “Observing Figure 3, it can be argued that the year 2015 comprehended studies that focused on perception, impact and social cost of EVs.” It was also explained in Discussion that authors allow the possibility that different social dimensions were covered in other papers that were excluded from this study (e.g. papers published in journals not listed in WoS, conference papers etc.”
Round 2
Reviewer 2 Report
The paper has improved after the revision. The number of articles still remains small, which is a weakness. At least the claims have been adjusted to reflect this small number of data points.
Please make sure you remove the "red-font" from the changes made in the final version.